# Can one hear the full nonlinearity of a PDE from its small excitations?

**Maxim Olshanii and Danshyl Boodhoo**⋆

Department of Physics, University of Massachusetts Boston,
Boston Massachusetts 02125, USA

⋆ danshyl.boodhoo@umb.edu

## Abstract

In this article, we show how one can restore an *unknown* nonlinear partial differential equation of a sine-Gordon type from its linearization around an unknown stationary kink. The key idea is to regard the ground state of the linear problem as the translation-related Goldstone mode of the nonlinear PDE sought after.

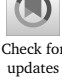

## 1  Introduction

In mathematical physics, inverse problems are ones of the hardest. The most famous one goes by the name "Can One Hear the Shape of a Drum?" [1]. There, one attempts to infer the shape of a membrane from the spectrum of its vibrations. Identifying a quantum scatterer from its scattering data [2] is another example. In this article we pose the following question: "Given a linear stability analysis equation around an unknown stationary solution of an unknown nonlinear Partial Differential Equation (PDE), can one restore the PDE itself?" Below, we will be using the term "inverse linearization" for such a procedure.

## 2  Principal result: The inverse linearization protocol for PDEs of the sine-Gordon type

Consider a linear stationary Schrödinger equation:

$$-\delta u_{xx} + V(x)\delta u = E\delta u, \tag{1}$$

with $\delta u = \delta u(x)$. Assume that the potential $V(x)$ supports at least one bound state. Consider its ground state $\delta u_0(x)$, of an energy $E_0$. It is known that the ground state $\delta u_0(x)$ does not have nodes and that it can be made purely real given an appropriate prefactor [3]. Accordingly, from now on, we will assume that the ground state obeys

$$\delta u_0 \overset{-\infty < x < +\infty}{>} 0.$$

Let us now rewrite the Schrödinger equation (1) as

$$-\delta u_{xx} + (V(x) - E_0)\delta u = \omega^2 \delta u, \tag{2}$$

with $\omega^2 \equiv E - E_0$. (Obviously, the ground state corresponds to $\omega = 0$; it obeys $-(\delta u_0)_{xx} + (V(x) - E_0)\delta u_0 = 0$.) Imagine now that the equation (2) constitutes an eigenmode equation for small excitations around a stationary kink-soltion of an unknown PDE of the sine-Gordon type, further referred as a blank-Gordon equation:

$$u_{xx} - u_{tt} = F(u), \tag{3}$$
$$u = u(x, t).$$

It is assumed that $F(u)$ is smooth. We claim the following:

1. The kink of (3) is related to the ground state of (1) as

$$\bar{u}(x) = A\left(B + \int_{-\infty}^{x} \delta u_0(x')\,dx'\right), \tag{4}$$

    with $A$ and $B$ being arbitrary constants;

2. The nonlinearity $F(u)$ is given by

$$F(u) = f(\chi(u)), \tag{5}$$

    where $f(x)$ is proportional to the first derivative of the ground state of (1),

$$f(x) = A(\delta u_0)_x(x), \tag{6}$$

    and the function $\chi(u)$ is the inverse of $\bar{u}(x)$:

$$\chi(\bar{u}(x)) = x.$$

The rigorous formulation of the time-dependent small excitation problem is as follows. The small excitations dress a stationary kink $\bar{u}(x)$,

$$\bar{u}_{xx} = F(\bar{u}). \tag{7}$$

A general time-dependent solution $u(x, t)$ of the nonlinear PDE (3) is represented as

$$u(x, t) = \bar{u}(x) + \delta u(x)e^{\pm i\omega t} + \mathcal{O}(\delta u(x)^2). \tag{8}$$

The expansion (8) is substituted in the nonlinear equation (3), and subsequently, only the terms of the first order in $\delta u(x)$ are kept. This procedure results in the following eigenmode equation:

$$-\delta u_{xx} + F'(\bar{u}(x))\delta u = \omega^2 \delta u. \tag{9}$$

## 3 Justification of the inverse linearization procedure (4), (5), (6)

First of all, in Section 2, we assumed that the inverse $\chi(\bar{u})$ of $\bar{u}(x)$ (see (4)) exists. Thanks to the positivity of $\delta u_0(x)$, the existence of this inverse is, however, guaranteed: $\bar{u}(x)$ is a monotonic function of $x$.

Secondly, we need to prove that

$$F'(\bar{u}(x)) = V(x) - E_0. \tag{10}$$

Indeed

$$F'(\bar{u}(x)) = \frac{d}{du}F(u)\Big|_{u=\bar{u}(x)} \tag{11}$$

$$= \left(\frac{d}{dx}f(x)\Big|_{x=\chi(\bar{u}(x))}\right)\left(\frac{d}{du}\chi(u)\Big|_{u=\bar{u}(x)}\right) \tag{12}$$

$$= \frac{\frac{d}{dx}f(x)}{\frac{d}{dx}\bar{u}(x)} \tag{13}$$

$$= \frac{A(\delta u_0)_{xx}(x)}{A\delta u_0(x)} \tag{14}$$

$$= V(x) - E_0. \tag{15}$$

End of proof.

However, the proof above tells one nothing about *why* the recipe works. The cornerstone for *understanding* the physics behind the connection formulas (4,5, 6) is the Goldstone theorem: if a PDE possesses a continuous symmetry, the action of the generator of this symmetry on a stationary solution is a small excitation around this solution, with a zero frequency. In our case, the symmetry in question is the translational invariance of the PDE (3) in question. Therefore the spatial derivative of a stationary solution is a legitimate zero-frequency mode of the linear stability analysis equation around that stationary solution. Conversely, the kink is proportional to an integral over that eigenstate; hence the connection (4), where it is assumed that it is the ground state of (1) that corresponds to the Goldstone mode. This is the connection between a nonlinear PDE and its linearization that we are exploiting in this work. This connection can also be seen explicitly by differentiating (7) with respect to $x$ and observing that the result is identical to the linearization (9) with $\omega = 0$, applied to the kink state $\bar{u}$.

## 4   Worked examples

First of all, we would like to verify that our procedure returns the sine-Gordon equation, if applied to its linearization around its stationary soliton. In that case the potential of the linearized problem is the Pöschl-Teller potential [4], $V(x) = -n(n+1)\operatorname{sech}^2(x)$, with $n = 1$ (see row 1 of Table 1). Using its ground state, $\delta u_0 = \operatorname{sech}(x)$, to initialize the protocol in the protocol (4), (5), (6), we obtain, by applying (4):

$$\bar{u}(x) = A(B + 2\arctan e^x). \tag{16}$$

Define the function:

$$u(x) = \frac{A}{2}\sin\left(\frac{2\bar{u}(x)}{A} - 2B\right). \tag{17}$$

Then following (5) and (6), the nonlinearity in the blank-Gordon equation $F(\bar{u})$) is given by:

$$F(\bar{u}) = \frac{A}{2}\sin\left(\frac{2}{A}\bar{u} - 2B\right) = \sin(u). \tag{18}$$

For $A = 2, B = 0$ we recover sine-Gordon equation. For arbitrary $A$ and $B$, the linear change of variables $\bar{u} \to u$, given by (17) yields the sine-Gordon equation in $u$, so the constants $A$ and $B$ do not affect the dynamics.

Table 1: Worked examples of the application of the inverse linearization procedure that allows to restore an unknown PDE from its linear stability analysis counterpart, around a stationary kink soliton. The *ground state* of the linear system (a linear Schrödinger equation in our case) is the translational Goldstone mode of the problem. The first column corresponds to the potential of the linear problem (1), with $E_0$ (second column) being its ground state energy. The third and the forth columns are the parameters of the ground state vs. kink connection (4). The fifth column identifies the relevant kink soliton. The sixth column gives the revealed PDE. Here and below, $\varphi(u) \equiv \sin\left(\frac{\arcsin(u)}{3}\right)$. The function $\mathrm{inverf}(u)$ is the inverse of the error function.

| $V(x)$ | $E_0$ | $A$ | $B$ | $\bar{u}(x)$ | $F(u)$ |
|---|---|---|---|---|---|
| $-2\,\mathrm{sech}^2(x)$ | $-1$ | $2$ | $0$ | $4\arctan(e^x)$ | $\sin(u)$ (sine-Gordon) |
| $-6\,\mathrm{sech}^2(x)$ | $-4$ | $1$ | $-1$ | $\tanh(x)$ | $-2u+2u^3$ ($\phi^4$ model) |
| $-12\,\mathrm{sech}^2(x)$ | $-9$ | $1$ | $0$ | $\arctan(e^x)+\frac{\sinh(x)}{2\cosh^2(x)}$ | no elementary functions for $F(u)$ |
| $-20\,\mathrm{sech}^2(x)$ | $-16$ | $\frac{3}{2}$ | $-\frac{2}{3}$ | $\frac{3}{2}\tanh(x)-\frac{1}{2}\tanh^2(x)$ | $-12\varphi(u)\left(1-4\varphi^2(u)\right)$ |
| $-\delta(x)$ | $-\frac{1}{4}$ | $\frac{1}{2}$ | $-2$ | $\mathrm{sign}(x)(1-e^{-|x|/2})$ | $\frac{1}{4}(u-\mathrm{sign}(u))$ (sawtooth-Gordon) |
| $4x^2$ | $2$ | $\frac{2}{\sqrt{\pi}}$ | $-\frac{\sqrt{\pi}}{2}$ | $\mathrm{erf}(x)$ | $-\frac{4e^{-\mathrm{inverf}(u)^2}\,\mathrm{inverf}(u)}{\sqrt{\pi}}$ |

In fact the constants $A$ and $B$ do not affect the dynamics, as solutions that differ only through the choice of $A$ and $B$ are related through an affine transformation. Suppose $u(x)$ is a stationary solution to the blank-Gordon equation with nonlinearity $F(u)$. That is $u''(x) = F(u)$. Suppose that:

$$v(x) = A(u(x)+B)u(x) = \frac{v(x)-B}{A} \,. \tag{19}$$

Both $u(x), v(x)$ can be obtained from (4), corresponding to an arbitrary choice of $A, B$.

Then:

$$v'' = \bar{F}(v) := AF\left(\frac{v(x)-B}{A}\right) + B \,.$$

Then changing variables from $v \to u$ yields the equation $u'' = F(u)$, so that the dynamics are the same regardless of the choice of A and B. Since $A, B$ do not affect the dynamics, the examples shown in the table use values $A, B$ that yield well-known or particularly simple $F(u)$.

For $n = 2$, a particular instance of the $\phi^4$ model is produced (row 2 of the Table 1). The PDE produced by the $n = 3$ Hamiltonian (row 3) does not appear to be expressible through elementary functions. The $n = 4$ case (row 4) appears to produce a PDE that has not been studied before. Note however that in this case, the function $F(u)$ has branching points at $u = \pm 1$. It is not clear if the corresponding PDE will be well defined there.

The $\delta$-well and the harmonic oscillator case are also included in our study (rows 5 and 6 respectively). In the former case, the nonlinearity $F(u)$ is a piece-wise-linear, periodic function of $u$: we named the resulting PDE a sawtooth-Gordon equation [5]. This form of $F(u)$ runs against our assumption of smooth nonlinearity, but our scheme still holds in the sense of distributions in this case. In the latter case, a function $F(u)$ that is expressed in terms of the inverse of the error function emerges. Similarly to the $n = 4$ Pöschl-Teller case, harmonic oscillator leads to a nonanalytic PDE.

# 5 Conclusions and outlook

In this article, we showed that information about low amplitude excitations of a sine-Gordon-type PDE with an unknown nonlinearity (the blank-Gordon equation), in the vicinity of its single kink stationary state (also unknown), is sufficient to recover that nonlinearity. If one regards the blank-Gordon equation as a toy field theory, our finding would also imply that in a two-vacuum state of that theory, low-energy physics contains full information about the high-energy sector.

Next, in [5], it has been shown that integrable PDEs produce reflectionless linear stability analysis equations. Regretfully, the converse is not true, as witnessed by row 2 of Table 1 pertaining to the $n = 2$ Pöschl-Teller Hamiltonian (see [6] on its transparency) and a particular $\phi^4$ model ($F(u) = -2u + 2u^3$) whose non-integrable behavior has been well documented [7].

Looking ahead, several directions of future research can be foreseen. First note that the ground state of the linear Schrödinger equation (1) is not the only eigenstate that produces a monotonic (and hence invertible) kink state $\bar{u}(x)$ (see (4)). For $V(x)$ supporting a single bound state or no bound states at all, the $E = 0$ eigenstate can be chosen to be monotonic, leading to a monotonic $\bar{u}(x)$. In that case the equation (2) needs to be modified as

$$-\delta u_{xx} + V(x)\delta u = \omega^2 \delta u\,, \tag{20}$$

with $\omega^2 \equiv E$. For example, the $E = 0$ of the $n = 1$ Pöschl-Teller potential produces the Liouville equation [5,8], whose kink solitons grow linearly at $x \to \pm\infty$. Remark that the latter behavior is not generic: it follows from the absence of a linear divergence of the $E = 0$ eigenstate of the $n = 1$ Pöschl-Teller Hamiltonian, the former being specific to the quantum supersymmetric Hamiltonians [9].

In retrospect, the monotonicity of the kink state $\bar{u}(x)$ is a sufficient but not a necessary condition for our inverse linearization scheme to function. As follows from the equation (7), any function $\bar{u}(x)$ whose second spatial derivative $\bar{u}_{xx}(x)$ is a function of $\bar{u}(x)$ itself can be a candidate for a kink soliton of a nonlinear PDE. And purely monotonic functions $\bar{u}(x)$ are not the only members of this class. It can be proven that there exist three and only three cases when

$$\bar{u}_{xx}(x) = F(\bar{u}(x)) : \tag{21}$$

Case 1. $\bar{u}(x)$ is monotonic.

Case 2. $\bar{u}(x)$ has a single extremum and it is spatially even with respect to this extremum.

Case 3. $\bar{u}(x)$ is periodic; $\bar{u}(x)$ has a single local maximum and a single local minumum per period, the former being separated by a half-period; $\bar{u}(x)$ is spatially even with respect to each of its extrema.

Case 1 is obvious. To prove that the remaining instances are covered by the Cases 2 and 3, observe the following. The equation (21) can be regarded as a second order ODE. At a point of extremum, $x_0$, the initial conditions on $\bar{u}(x_0)$ and $\bar{u}_x(x_0) = 0$ are spatially even, and they will generate a function that is even about $x_0$. If this is the only extremum, the Case 2 is covered. In case if there are more than one extrema, it is easy to show inductively that any pair consisting of a minimum and a maximum, at $x_{0,\,\mathrm{min}}$ and $x_{0,\,\mathrm{max}}$, with no extrema in between generates a periodic function with a period of $2|x_{0,\,\mathrm{max}} - x_{0,\,\mathrm{min}}|$. This constitutes the Case 3.

As an example of an inverse linearization pertaining to the Case 2, one may consider the first excited state $\delta u_1(x)$ (of energy $E_1 = -1$) of the $n = 2$ Pöschl-Teller problem leading to a different kind of the $\phi^4$ model, with $F(u) = u - 2u^3$. The analogue of the equation (20) will read

$$-\delta u_{xx} + (V(x) - E_1)\delta u = \omega^2 \delta u\,, \tag{22}$$

with $\omega^2 \equiv E - E_1$.

We were not able to identify any instances corresponding to Case 3.

It should be noted that whenever the Goldstone mode is not represented by the ground state of the linearization problem, the corresponding kink soliton is dynamically unstable. Indeed, in that case, the ground state, as well as any other eigenstate with energy below the Goldstone value, will have an imaginary frequency.

One more comment is in order. Rigorously speaking, the connection (4), (5), (6) generates a *family* of PDEs. While the constant $A$ trivially corresponds to normalization of $\delta u_0$ (more generally, the state associated with the Goldstone mode), different values of the constant $B$ may produce different PDEs. This freedom needs to be explored further.

## Acknowledgments

We are grateful to Vanja Dunjko for his valuable comments.

**Funding information** MO was supported by the NSF Grant No. PHY-2309271.

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
