# Peer review of "Can one hear the full nonlinearity of a PDE from its small excitations?"

_SciPost Physics Core, doi:SciPost Phys. Core 8, 055 (2025)_

## Round 1 · Referee Report · Anonymous (Referee 1) · 2024-12-16

Report

This is an interesting article which flips the usual connection between a kink like state in a translation invariant system and the corresponding eigenfunction, its derivative, with eigenvalue zero (sometimes called the Goldstone mode). After performing what seems to be a sound derivation, the authors provide several examples in Klein-Gordon type examples where explicit kink-like solutions are known. I think it is a worthwhile idea to share in such a venue but I have a few comments and one question which could clarify and improve the article.

  1. The inverse $\chi$ for $\bar u$ is only defined on $u$ values in the range of $\bar u$. How would this allow one to obtain the values of F(\bar u) for \bar u outside this range? While I do not doubt the authors worked out examples, I wonder how, in general, one can obtain information about the nonlinear system outside this range. Namely, could F be defined differently outside this range and not affect the inverse problem? Certainly something the authors should discuss. This also connects to the authors comment Sec. 5 that this idea indicates that low energy dynamics indicate the higher energy dynamics.

  2. The authors give some nice discussion in Sec. 5, but I think more discussion on the use of this idea would be helpful to the reader. Namely, is there an example of a known kink-like coherent structure and associated linear equation where the associated nonlinear system is not known? Further, where do the authors expect this to be useful/helpful? Is there an example where one has an (approximate) understanding of some ground state eigenfunction (either numerically or experimentally) but not the underlying nonlinear equation?

  3. This seems to work only when the ground state is algebraically simple. Are there relevant cases (maybe for a different linear operator) where degenerate zero eigenvalues occur? This might possibly preclude the analysis done here.

  4. How much of this analysis do the authors expect to be able to be lifted to higher spatial dimensions (where there are more translation invariant norms? If the kink is localized in multiple directions, then I would expect multiple 0-eigenvalues to exist.

A few small typos: Pg. 3, below the proof: “above tell one nothing” to “above tells one nothing” Pg. 4, above (16): “modifies” to “modified”

Recommendation

Ask for minor revision

  • validity: -
  • significance: -
  • originality: -
  • clarity: -
  • formatting: -
  • grammar: -

Author:  Danshyl Boodhoo  on 2025-03-12  [id 5284]

(in reply to Report 1 on 2024-12-16)

Thank you for your report.

  1. For general $\bar{u}$ it appears to be the case that the domain of $F$ may be extended in more than one way. However, if $\bar{u}$ is analytic, then it can be shown that $F$ must be analytic as well on the range of $\bar{u}$. Analyticity guarantees a unique extension of $F$.

  2. The main hope is to ultimately apply this technique to determine high-energy behavior of fields from their low energy excitations. Such a case may be found, for example, in the behavior of W-boson molecules. In table 1 of the paper, we invert a simple harmonic potential resulting in an error function solution. The authors are unaware of a corresponding well-understood nonlinear problem. The authors are also unaware of any other linearization that would result, upon inversion in an elegant nonlinear PDE.

  3. The inverse linearization scheme depends on the eigenfunction, not just the eigenvalue. Therefore even if there is degeneracy in the system, the procedure should not change. That said, we do not know of any systems with degeneracy.

  4. The scheme relies on stationary soliton solutions, and not many are known in higher dimensions. Of the few known multidimensional stationary solutions, one has the Townes soliton, which may be amenable to the analysis in this paper, but is out of the scope of the paper.

Author:  Danshyl Boodhoo  on 2025-01-07  [id 5092]

(in reply to Report 1 on 2024-12-16)
Category:
answer to question

Thank you for your helpful comments.

  1. For general $\bar{u}$ it appears to be the case that the domain of $F$ may be extended in more than one way. However, if $\bar{u}$ is analytic, then it can be shown that $F$ must be analytic as well on the range of $\bar{u}$. Analyticity guarantees a unique extension of $F$.

  2. The authors are currently applying this technique to a tight-binding model. The ground state eigenfunctions are well-understood numerically for this system.

  3. For algebraically simple ground states it is possible to explicitly compute $F$, but the scheme only relies on $F(\bar{u})$ being well-defined. Therefore it may not be possible to explicitly compute the form of $F$, though it is still possible to define $F$ through this scheme. The inverse linearization relies on choosing a particular eigenfunction; different eigenfunctions may lead to different $F$. However, this is to be expected since different nonlinear equations can have the same linearization, so it shouldn't preclude this analysis.

  4. The main role of translational invariance in this scheme is to provide a continuous symmetry of the PDE that is guaranteed to exist regardless of the form of $F$. Each continuous symmetry corresponds to a different linearized equation. Therefore in higher dimensions it should still be possible to construct a scheme such as this one, and it may even be possible to extract more information about $F$ given several different linearizations, one for each symmetry. On the other hand, the establishment of what kind of eigenfunctions permit such an inverse linearization, as outlined in section 5, relies on the problem being one-dimensional. A different approach to establishing the solutions that permit inverse linearization may be needed in higher dimensions.

Thank you for pointing out the typos, these will be corrected.

---

## Round 1 · Referee Report · Anonymous (Referee 2) · 2025-1-1

Report

This paper explores a classical inverse problem for PDEs of general sine-Gordon type, $u_{xx}-u_{tt} = F(u)$ using a succinct construction relating a stationary (kink) solution to the ground state of an associated linear Schrödinger equation.

By appealing to the Goldstone Theorem, translational invariance of the PDE is used to write down an integral equation of the (stationary) kink solution involving the ground state; conditions on the nonlinearity of $F$ are also made explicit via a function of the kink and ground state.

I recommend this report for publication with only minor corrections, listed alongside some general comments to be considered, below.

Requested changes

The authors assume the potential $V(x)$ supports at least one bound state, as supported by the examples on Table 1. It would be valuable to know if there are any examples of physical interest for which this is *not* the case; any comments that could be made on the regularity or smoothness of the potential (and consequently the bound state) would be nice to add here, though not absolutely essential.

As the Goldstone Theorem relies on the existence of continuous symmetries of the PDE, it would be good to add a remark on the conditions on F, even if only for the explicit examples considered.

In Table 1, how were A and B determined? It does seem that there are multiple choices for these constants which could do the job and it may not be obvious whether the PDE produced will be immediately recognizable. This is mentioned in the final paragraph of the conclusion, so a brief remark on how the constants were found for these examples would be helpful to the reader.

When applying Goldstone's Theorem, the translational invariance of the PDE in (3) is used. It would be interesting to apply the Theorem to those equations in Table 1 which contain both translational and rotational invariance and compare the resulting integral equations; this could lead to a different understanding of the mechanics of the inverse problem.

In a similar vein, the principles of this paper could also be applied, for example, to elliptic equations, i.e., by changing the signature $u_{tt}-u_{xx}$ to $u_{tt}+u_{xx}$. At first glance, this would be resolved by a simple Wick rotation $t \rightarrow$ $it$, but the resulting elliptic equation could have interesting ramifications for the inverse problem over $\mathbb{C}$ instead of $\mathbb{R}$.

Typos
Page 3 - Last paragraph of section 3, 4 lines from the end of the paragraph – “Golstone” should be “Goldstone”
Page 4 – “Several direction OF future research can be foreseen”
Page 4 – Line above (16) “modifies” should be “modified”
Page 5 – Paragraph just before (18) “fist” should be “first”
Page 5 – Paragraph just after (18) “eigensate” should be “eigenstate”

Recommendation

Ask for minor revision

  • validity: -
  • significance: -
  • originality: -
  • clarity: -
  • formatting: -
  • grammar: -

Author:  Danshyl Boodhoo  on 2025-03-12  [id 5283]

(in reply to Report 2 on 2025-01-01)
Category:
question
answer to question
validation or rederivation

Thank you for your report.

Regarding conditions on V(x), the scope of the paper is limited to kink solitons. The Goldstone mode is a derivative, and is thus localized. Therefore, it is likely that we will be dealing with a bound state.

The authors will include a remark on differentiability of F and on the choices for constants A and B in the updated version.

Regarding your point on rotational invariance, we are not sure we understand. Since the system is 1 dimensional, there shouldn't be a rotation.

Finally, regarding elliptic versions, the PDE changes overall and the sign change changes the dynamics so that formerly stable excitations become unstable, and vice-versa. However, because we are considering linearization around stationary solutions, the inverse linearization scheme is the same regardless.

---

## Round 2 · Referee Report · Anonymous (Referee 1) · 2025-7-21

Report

The authors have addressed my comments and I believe it is suitable for publication.

Recommendation

Publish (meets expectations and criteria for this Journal)

---

## Round 2 · Referee Report · Anonymous (Referee 2) · 2025-7-21

Report

Thank you for the clear responses to the questions posed by both referees. Seeing these clarifications, I see that my comment regarding rotational invariance is not relevant to the problem being solved (or to a natural extensions in higher dimensions). The corrections made within the article address my questions directly while not detracting from the main conclusions of the article. I recommend the corrected version for publication.

Recommendation

Publish (meets expectations and criteria for this Journal)

---

## Round 2 · Author Response

This resubmission addresses comments from reviewers.

---

## Round 2 · List of Changes

• Added a regularity condition on the nonlinearity $F(u)$

  • Added a section on the relevance, or lack thereof, of the arbitrary constants A and B

  • Added an example calculation for the method that reproduces the sine-Gordon equation for a given choice of A and B.

---

## Editorial Decision

published